# Identification of Endometrial Cancer-Specific microRNA Biomarkers in Endometrial Fluid

**DOI:** 10.3390/ijms24108683

**Published:** 2023-05-12

**Authors:** Jianing Yang, Joel E. Barkley, Bikash Bhattarai, Kameron Firouzi, Bradley J. Monk, Dean V. Coonrod, Frederic Zenhausern

**Affiliations:** 1Center for Applied NanoBiosciences and Medicine, University of Arizona College of Medicine-Phoenix, Phoenix, AZ 85004, USA; fzenhaus@arizona.edu; 2Department of Obstetrics and Gynecology, District Medical Group, Valleywise Health, Phoenix, AZ 85008, USA; kfirooz7@gmail.com (K.F.); dean_coonrod@dmgaz.org (D.V.C.); 3Department of Obstetrics and Gynecology, Creighton University, Phoenix, AZ 85012, USA; bmonk@gog.org; 4Department of Obstetrics and Gynecology, University of Arizona College of Medicine-Phoenix, Phoenix, AZ 85004, USA; bikash@arizona.edu; 5Department of Research, Valleywise Health, Phoenix, AZ 85008, USA; 6HonorHealth Research Institute, Scottsdale, AZ 85258, USA; 7Department of Biomedical Engineering, University of Arizona’s College of Engineering, Tucson, AZ 85721, USA; 8Department of Basic Medical Sciences, University of Arizona College of Medicine-Phoenix, Phoenix, AZ 85004, USA

**Keywords:** endometrial cancer (EC), liquid biopsy, endometrial fluid, microRNA, saline infusion sonohysterography (SIS), endometrial biopsy, upregulation, downregulation, exploration, validation

## Abstract

Abnormal uterine bleeding is a common benign gynecological complaint and is also the most common symptom of endometrial cancer (EC). Although many microRNAs have been reported in endometrial carcinoma, most of them were identified from tumor tissues obtained at surgery or from cell lines cultured in laboratories. The objective of this study was to develop a method to detect EC-specific microRNA biomarkers from liquid biopsy samples to improve the early diagnosis of EC in women. Endometrial fluid samples were collected during patient-scheduled in-office visits or in the operating room prior to surgery using the same technique performed for saline infusion sonohysterography (SIS). The total RNA was extracted from the endometrial fluid specimens, followed by quantification, reverse transcription, and real-time PCR arrays. The study was conducted in two phases: exploratory phase I and validation phase II. In total, endometrial fluid samples from 82 patients were collected and processed, with 60 matched non-cancer versus endometrial carcinoma patients used in phase I and 22 in phase II. The 14 microRNA biomarkers, out of 84 miRNA candidates, with the greatest variation in expression from phase I, were selected to enter phase II validation and statistical analysis. Among them, three microRNAs had a consistent and substantial fold-change in upregulation (miR-429, miR-183-5p, and miR-146a-5p). Furthermore, four miRNAs (miR-378c, miR-4705, miR-1321, and miR-362-3p) were uniquely detected. This research elucidated the feasibility of the collection, quantification, and detection of miRNA from endometrial fluid with a minimally invasive procedure performed during a patient in-office visit. The screening of a larger set of clinical samples was necessary to validate these early detection biomarkers for endometrial cancer.

## 1. Introduction

The American Cancer Society recently published their estimations for uterine cancer in the United States for 2022, most of which were endometrial carcinomas. They estimated that there were 65,950 new cases of uterine cancer and 12,160 deaths attributed to the disease [1]. The perspectives in the literature on the impact of diagnosis delay in endometrial cancer are mixed [2]. A retrospective review in 2002 reported a paradoxically inverse relationship between surgical waiting times and survival [3]; however, a subsequently larger study showed that waiting times affected overall survival [4]. Direct studies of diagnosis delay are difficult to perform, even retrospectively, as the workup interval prior to diagnosis is frequently unreported and is subject to recall bias regarding symptom onset (abnormal bleeding) by patients. Hence, the establishment of a reliable method, with easily accessible, minimally invasive, and accurately detectable biomarkers, that can be used to screen the population with abnormal uterine bleeding (AUB) and improve the early detection and diagnosis of endometrial carcinoma is significant in endometrial cancer research.

The endometrium is the inner lining of the uterus. Endometrial cancer (EC) is the most common cancer of the female genital tract in the United States. The average chance of a woman being diagnosed with endometrial cancer during her lifetime is approximately 1 in 37 [1]. Endometrial cancer is usually detected at an early stage because it frequently presents with abnormal uterine bleeding, which prompts women to see their doctors. Abnormal uterine bleeding (AUB) is the reason for approximately 70% of gynecological consultations sought by perimenopausal women [5]. Many women who experience AUB are considered at high-risk for uterine cancer due to their age, body mass index, coexisting medical conditions, or the failure of initial medical therapy. In these high-risk groups, abnormal bleeding necessitates further evaluation to exclude the presence of atypical hyperplasia or early-stage endometrial cancer prior to initial medical or surgical intervention. Endometrial biopsy (EMB) has been the mainstay of endometrial evaluation. However, it has been shown to sample a limited portion of the endometrial surface with a sampling bias (occasionally missing significant cancerous tissue), and it can be discordant with the final pathological results [6,7]. The current methods used for early EC detection were recently reviewed by Shen et al. [8]. These minimally invasive approaches, including transvaginal ultrasonography (TVU), saline infusion sonohysterography (SIS), uterine lavage, and cervicovaginal fluid, are briefly summarized.

Currently, there is no standard screening test for the early detection of EC. Discovering new molecular biomarkers and developing/refining non-invasive tests have been the endeavors of biomedical researchers working to improve the early and accurate detection and diagnosis of EC. One of these approaches is to identify and validate biomarkers in liquid biopsy samples. Liquid biopsy is a relatively non-invasive procedure that can detect cancer-associated biomarkers, including different types of cancer of the female reproductive system, such as ovarian cancer and endometrial cancer, using peripheral blood, saliva, urine, and uterine lavage or cervicovaginal fluid. It has been the focus of research as an alternative sampling method to traditional tissue biopsies [9,10,11,12]. Endometrial cancer originates from the endometrium. Considering that the endometrial lining sheds regularly and that biomarkers associated with circulating shed membrane fragments have been detected in endometrial cancer [13], it would be reasonable to hypothesize that endometrial-cancer-specific biomarkers could be detected in a liquid biopsy sample such as endometrial fluid obtained from the endometrial cavity. Endometrial fluid is a potential source of diagnostic samples that may have a benefit over EMB in that its samples come from the whole endometrial cavity; thus, it will have a higher level of accuracy and is easier to collect. By contrast, EMB may lack accuracy due to the heterogeneity of the biopsied tissues and the limited sampling area.

MicroRNAs (miRNAs) are a class of naturally occurring small non-coding RNA molecules, approximately 21–25 nucleotides in length. miRNAs are regulatory RNA molecules with diverse cellular functions and pathological implications. They function as post-transcriptional gene expression modulators, repress genes, and are critical for carcinogenesis [14]. miRNAs were first described 30 years ago by Lee and colleagues [15], and the term miRNA was coined in 2001 [16]. In recent years, thousands of oncogenic miRNAs have been identified. They encode genes in the human genome and target approximately 60% of mammalian genes. Specific miRNAs are expressed in various tissues, and the dysregulation of miRNA expression has been reported in a variety of diseases and in carcinogenesis [17], where they act by modulating oncogenes and tumor suppressor genes. Differences in the patterns of miRNA expression between normal and cancerous tissue samples can be useful in diagnosis, prognosis, and providing guidance for treatment. There are two classes of miRNAs that are involved in the carcinogenesis of EC: oncogenic miRNAs (onco-miRs) and tumor suppressor miRNAs (tumor suppressor miRs). The former is often found to be upregulated in tumors, and the latter is downregulated in tumors [18]. The involvement of miRNA in tumor oncogenesis, invasion, and metastasis has been reported in EC [18,19,20,21,22,23,24,25,26,27]. Most of the EC-specific miRNA biomarkers reported to date have been identified using samples from blood [23], cancerous tissue [25], and cell lines [27], or formalin-fixed paraffin-embedded (FFPE) endometrial biopsy tissues [23,24,25,27,28,29,30], which were obtained either in a post-surgical procedure or post-diagnosis. Several miRNAs, for example, miR-21-5p, miR-34a, miR-106b, miR-181a, miR-103, miR-155-5p, miR-200a, miR-200b, miR-429, and miR-7, are upregulated in EC and involved in oncogenesis, invasion, and metastasis [18,23,24,25,26]. However, miR-34b, miR-221, miR-152, miR-204, and miR-149 are downregulated in EC [23,24,25]. miRNA biomarkers have also been identified in a wide range of body fluids (plasma, plural fluid, saliva, urine, breast milk, etc.) [21,31,32]. In recent years, more and more miRNAs have been reported in endometrial fluids [33,34,35,36,37,38] sampled from cervicovaginal lavage, the implantation endometrium during embryo transfer, or uterine tissue during the secretory phase of the menstrual cycle. These reports were aimed at cervical cancer diagnosis and an endometrial micro-environment assessment for implantation receptivity. Only a few studies were available in the literature, suggesting that the molecular markers of EC, either DNA or protein markers, could be detected in lower genital tract lavage specimens [12,21,39]. However, no study has been performed to directly detect miRNA biomarkers in endometrial fluid for EC detection. The methods used for the miRNA profiling of the endometrial fluid have included exosome isolation, a luciferase activity assay, total RNA extraction, and cDNA synthesis followed by qPCR and next-generation sequencing (NGS) after total RNA isolation. The upregulation of miR-21 and miR-146a was identified to be associated with a high level of cervical-cancer-derived exosomes [33] in cervicovaginal lavage fluid. Furthermore, several miRNAs were detectable in non-cancerous uterine aspirates, such as miR-200b, miR200c-3p, miR-30d-5p, miR-92a-3p, miR-17-5p, miR-451a, miR-106a, etc. [34,37,38]. It has been shown that most miRNAs are present inside cells [40]. Exosomes that contain miRNAs were released from endometrial epithelial cells into the uterine fluid, and the tetraspanins CD9 and CD63 of exosome markers were detectable on the apical surface of endometrial epithelial cells [38] using immunostaining. The endometrium proliferates and sheds cyclically under hormonal influence [19]. The biogenesis process of endometrial miRNAs depends on the endometrial cells’ physiopathological conditions, their microenvironment, and other factors [23]. Taking all aspects into consideration, it is logical to hypothesize and propose that, for early EC detection, the practical way to identify EC-specific miRNAs is to isolate them from endometrial epithelial cells shed in endometrial fluid.

Saline infusion sonohysterography (SIS) is an in-office procedure for endometrial evaluation [8] and is often performed on gynecological patients who experience abnormal uterine bleeding. During SIS, a sterile saline solution is instilled into the uterine cavity trans-cervically to provide a contrast when a transvaginal ultrasound examination is performed and improves the diagnosis of endometrial diseases. This exam can be conducted in a doctor’s office or clinic. Routinely, the injected saline solution is allowed to spontaneously expel at the end of the procedure. We collected the uterine aspirate of both control (non-EC) and EC patients and used this endometrial fluid for miRNA isolation and profiling in the current study.

The objective of this project was to perform a proof-of-concept study and develop a minimally invasive method to collect endometrial fluid during a patient in-office visit and to identify and validate miRNA biomarkers that could differentiate between benign and cancerous endometrial conditions.

## 2. Results

Endometrial fluid sample collection occurred upon patient consent, either in the clinic or operating room. The process of endometrial fluid sample collection followed the guideline of fluid biospecimen collection, processing, and storage in endometriosis research [39].

### 2.1. Study Design and Workflow

Endometrial fluid samples were collected from 82 patients (42 cancer and 40 controls). They were processed using 60 paired patient samples, i.e., 30 non-cancer and 30 EC samples, in phase I, and 22 patient samples, i.e., 10 non-cancer and 12 EC samples, in phase II of the study. The fourteen microRNA biomarkers with the greatest variation (>2-fold change) in expression levels from phase I were used for validation and comprehensive statistical analysis.

The physical properties, volume, and color of the collected endometrial fluid specimens were carefully recorded (Appendix A). Bloody fluid and tissue strings were observed in some collected samples, which led to repeated measurements during data analysis.

To increase the yield of each isolation, a duplicate extraction of 200 μL was performed for each patient SIS sample, and these were combined during the step of RNA capture in the MinElute column. The schematic diagram below shows the whole workflow process from SIS endometrial fluid collection to data analysis (Figure 1).

### 2.2. miRNA Extraction and Selection

The total RNA, including miRNA, was extracted from endometrial fluid specimens. The average quantity of the total RNA, including miRNA, extracted from the endometrial fluid in both phase I and phase II ranged from 47 ng/μL to 80 ng/μL, with substantial standard deviations (Appendix A). This was mainly due to the variations in the properties and quality of the collected samples. The appearance of the endometrial fluid samples collected from patients varied in color (from clear, yellowish, pink, to bloody), viscosity (from watery to viscous), and amount of tissue debris, which could affect the quantity and quality of RNA isolated from endometrial fluid samples. The RNA extraction was repeated if the RNA sample integrity was poor, as evaluated using an Agilent Bioanalyzer (RIN < 5), and/or if inhibition was suspected during downstream assays.

During phase I, 35 miRNAs out of the 84 miRNAs selected was expressed in the endometrial fluid of EC patients, exhibiting a >1-fold change (Figure 2). There were 19 miRNAs with significantly upregulated expression levels (>2-fold increase), and 6 miRNAs were significantly downregulated (>2-fold decrease) compared to the non-cancer controls (Table 1), accounting for 54.3% and 17.1% of all miRNAs identified in the EC endometrial fluid samples, respectively. Among them, the expression levels of hsa-miR-183-5p, hsa-miR-429, and hsa-miR-182-5p showed more than a 4-fold increase in the upregulation detected in EC samples compared with the non-EC samples, while the expression of hsa-miR-204-5p exhibited a dramatic downregulation (>4-fold decrease). There were several miRNA species detected in the EC endometrial fluid that have not been reported previously in EC (i.e., were not matched in the published list of miRNAs identified in EC). These were miR-362-3p, miR-378c, miR-4705, and miR-1321. Additionally, the downregulation of miR-378c, miR-4705, and miR-1321 and the upregulation of miR-362-3p were identified in EC endometrial fluid samples in the exploration phase (phase I).

A detailed statistical analysis of fold-changes, with *p*-values, confidence ranges, and effect sizes, can be found in the Appendix A.

### 2.3. Phase II—Validation of 14 miRNAs

A total of 14 miRNA species with substantial fold-changes detected in EC patient samples in phase I of the study were selected for phase II of the study (the validation phase; Table 2). An additional 22 patients’ endometrial fluid samples were collected, including 12 EC patients and 10 non-EC control patients. Multiple runs of each patient specimen were considered a “group”, and there were 12 cancer sample groups and 10 control sample groups. A minimum of two duplicate miScript miRNA PCR arrays were performed for every patient’s SIS endometrial fluid sample, and the repeat runs were performed on different miScript PCR array plates to account for possible pipetting and plate variations. Fold-changes relative to the plate control (ΔCT) and then fold-changes in miRNA detected in EC patient endometrial fluid (ΔCT) were compared with the controls (ΔCT) on the same plate (ΔΔCT). The final comparison between the average of all EC patients (ΔΔCT) and the average of all non-EC control patients (ΔΔCT) was calculated (2^−ΔΔCT^) and is presented in Figure 3.

The correlation of the exploration-phase fold-change values across miRNAs with those expressed by EC specimens from the validation phase was measured using Spearman’s rho. The observed correlation coefficients ranged from a strong correlation (rho = 0.78, *p* < 0.001) to a mild negative correlation of rho = −0.41 (*p* = 0.141). For the upregulated miRNAs, a strong correlation in certain miRNAs expressed in EC specimens was observed between the validation phase (phase II) and the exploration phase (phase I); for example, the levels of upregulation for miR-183-5p, miR-429, and miR-146a-5p were consistent in both phases (Table 2). The fold-changes in miR-429 were 4.3 and 4.59 in phase I and phase II, respectively. The fold-changes in expression levels for miR-183-5p were 4.75 in phase I and 3.79 in phase II. The fold-changes for miR-146a-5p from the EC patient endometrial fluid samples were 2.37 and 2.19 in phase I and phase II, respectively. Three out of ten upregulated miRNAs (33%) exhibited only moderate fold-changes in the validation phase, with 1.34-, 1.45-, and 1.10-fold changes for miR-182-5p, miR-34a-5p, and miR-181a-5p, respectively. There were two miRNAs, i.e., miR-7-5p and miR-21-5p, that were observed with significant upregulations in phase I (3.3-fold and 2.26-fold) but showed no fold-changes for the upregulation in phase II. Instead, a mild downregulation was detected in phase II (−1.14-fold and −1.5-fold). Interestingly, two out of four downregulated miRNAs, i.e., miR-378c and miR-34b-3p, exhibited slight to moderate upregulations, which was opposite to the fold-changes observed in phase I (Table 2 and Figure 3).

Receiver operating characteristic (ROC) curves of fold-change in the Ct values of the three most prominent miRNAs were constructed to discriminate between benign and cancer. The areas under the fitted curves (AUC) were 0.675, 0.709, and 0.685 for miR-183-5p, miR-429, and miR-146a-5p, respectively (Figure 4).

These ROC curves demonstrate the ability of the test (here, ΔΔΔCt) to discriminate between cancer vs. non-cancer patients. A value above 0.5 indicated that the test had at least some diagnostic potential, with an ideal ROC of one indicative of perfect discrimination between cancer and non-cancer patients. The accuracy of the diagnostic value for the three prominent miRNAs, i.e., miR-183-5p, miR-429, and miR-146a-5p, is visualized in Figure 4. The sensitivity and specificity represented by the area under the ROC curves (AUROC) were calculated. The fold-change in Ct values (ΔΔΔCt) at various levels of each miRNA obtained from endometrial fluid specimens was plotted in terms of differentiating between the benign and cancer histopathology of the endometrium. Based on the AUROC values above 0.675 for the three miRNAs, it appeared that the fold-change values of these biomarkers could be useful in distinguishing between benign and cancer in patients at risk of endometrial cancer.

### 2.4. Clinical and Demographic Features of Patients in the Study

Eighty-two patients were successfully recruited, and sample collection was performed prior to surgery. The demographic patient features were recorded and compared between the EC group and the non-cancer control group (Table 3). Several standard categorical variables were compared, including race, ethnicity, contraception usage, smoking status, sexually transmitted infection (STI), age group, body mass index (BMI), gravidity, and parity.

Systematic differences between the two groups of non-EC and EC patients were compared in their clinical and demographic features. Among all categorical and continuous variables, it appeared that post-menopause status and age were clearly confounding factors in EC patients. Body mass index (BMI) may have also played an important role in EC occurrence. The histopathological classification of the EC samples is summarized in Table 4 for both phase I and phase II of the study.

A total of 42 endometrial carcinoma patients were enrolled in this study: 30 EC patients in phase I and 12 EC patients in phase II. Among the 30 EC patients enrolled in phase I, there was twenty-five with a final histology of endometrioid, four with endometrial intraepithelial neoplasia (EIN), and one with rhabdomyosarcoma. However, all 12 EC patients in phase II had endometrioid carcinoma. The tumor grade was also different, with the majority of ECs represented as grade two cancers in phase I (53%) and the majority of EC specimens presenting as grade 1 (83%) cancers in phase II.

## 3. Discussion

The aim of this study was to use a minimally invasive technique similar to SIS to collect endometrial fluid samples and identify EC-specific miRNA biomarkers for early diagnosis. The standard evaluation of AUB frequently includes in-office procedures, such as endometrial biopsy, transvaginal ultrasound, and saline infusion sonohysterography (SIS). SIS is a specialized type of minimally invasive in-office ultrasound procedure, during which saline is injected into the endometrial cavity to better characterize its features. At the end of the SIS procedure, the installed saline solution is normally expelled and discarded. In this study, we collected the post-SIS saline solution and used this liquid biopsy sample to screen for miRNA biomarkers.

In this study, a total of 84 candidate miRNAs were selected through a thorough search of the literature from previously published studies of miRNA species identified in EC [11,18,19,20,21,22,26,28,35,40] and miRNA target protein markers reported in uterine fluid [21,39]. Some of the candidate miRNAs were chosen based on their target protein biomarkers identified in EC, e.g., PTEN, p53, K-ras, HER2, P4HB, ACAA1, etc. [40,41,42,43,44], and their corresponding miRNAs could be determined using an online search engine [45]. As a result of the limited number of EC-specific miRNAs reported at the time of the study design, additional miRNA markers were selected from cancers derived from the female reproductive system (ovary, uterine, and cervical cancers) and were included in the panel of 84 candidate miRNAs used to screen and evaluate endometrial fluid specimens in phase I (the exploration phase) of the study.

Our results demonstrate that miRNA could be extracted and quantified from the endometrial fluid specimens collected using a technique such as SIS. A wide variation in RNA quantification was observed from the endometrial fluid specimens. The main contributor to the high STDV was the physiological properties of individual specimens received, as shown in Appendix A, which could be influenced by multiple factors, such as tissue burden, active bleeding, other existing diseases, age, and other confounding factors. The main contaminants were from non-tumor sources. The PCR inhibition encountered occurred about 6% of the time in our 82 patient specimens, and usually repeated miRNA isolation with another aliquot of endometrial fluid from the same patient would resolve this issue. Subsequently, variations in the expression for each individual miRNA species were recalculated.

### 3.1. EC-Relevant miRNAs Identified in Phase I of the Study

In phase I of the study, several miRNAs from our 84-candidate miRNA panel showed significant changes in their expression levels when detected in EC samples compared to the controls. For example, miR-183-5p, miR-429, and miR-182-5p exhibited an over 4-fold upregulation, and miR-204-5p showed an over 4-fold downregulation. These findings were consistent with reports in the literature on miRNAs identified in EC tumor tissues [18,20,24,25,26,27]. Other miRNA species that were upregulated, such as miR-7-5p, miR-200a-3p, miR-200b-3p, miR-34a-5p, and miR-19b-3p, or downregulated, such as miR-34b-3p, miR-1247-3p, and miR-152-3p, all exhibited a >2-fold increase or decrease in their expression levels when detected in EC endometrial fluid specimens, which are also aligned with the previous findings reported for EC [18,19,20,21,22,23,24,25,26]. Inconsistently, however, miR-296-5p was found to be downregulated in our study (>2-fold decrease) but was reported to be upregulated by Yoneyama et al. [27]. Since both studies used similar methods of miRNA profiling, i.e., the reverse-transcription and real-time polymerase chain reaction (RT-qPCR), followed by array-based miRNA expression analysis, the only major difference was the sample types. The samples used in their study were from EC tissue specimens or an endometrioid endometrial carcinoma (EEC)-derived cell line [27], while we extracted the RNA from the endometrial fluid collected prior to surgery, essentially a liquid biopsy sample containing miRNAs mainly from the endometrial surface after shedding. In addition, the histology, grade, and stage of EEC tissue specimens may have differed from our EC endometrial fluid samples.

Conflicting results for miRNA expression have also been reported for biopsied EC tissue or FFPE samples. For example, miR-34a was noted to be downregulated in EC tissue in one report [26] but upregulated in EC tissue in another [25]. Different results for the same miRNA (miR-34a) identified in the same sample types (biopsied human EC lesion or tissue) indicated variations in EC sampling, which is intrinsic to tissue biopsy. The biopsied EC tissue could contain a heterogeneous tumor cell population, and the cellular signaling processes could be differentially regulated, depending on the biopsy sites and classification of the EC lesion when biopsied at the time of sample collection. In the current study, we showed that miR-34a-5p was significantly upregulated in EC endometrial fluid, which is consistent with Favier et al. [25].

MicroRNAs modulate tumor progression, invasion, and metastasis by targeting various mRNAs via different signaling pathways [26]. The functional pathways of some miRNAs identified in EC endometrial fluid specimens have been reported in the literature. The overexpression of miR-183-5p led to the downregulation of the mRNA and protein expression of ezrin in EC cells [26,46], while the upregulation of miR-429 resulted in the activation of the transcription factor AP-2α in EC tissues [18,19] and the promotion of epithelial–mesenchymal transition (EMT) [21], but the downregulation of PTEN mRNA expression [27]. It is worth emphasizing that miR-146a-5p has been found in various cancers [47], but it has not been previously reported in EC tissue specimens or in uterine lavage, even though miR-146a was identified in tumor-associated macrophages (TAM) isolated from fresh EC tissue [47,48]. A higher level of miR-146a-5p was associated with high-grade cancer and tumor invasion [47,48]. Certain signaling pathways were differentially targeted by miR-146a-5p in various cancers, including LIF-Stat3 when reported in the endometrium of patients with implantation failure [23], NOTCH 1/2 in endometrial carcinoma [47], and the EGFR, NF-κB, TGF-β, and SMAD4 pathways in other tumor types. Our results are the first to report the upregulation of miR-146a-5p in EC endometrial fluid, although it was not denoted as “unmatched” miRNAs (Table 1) in this study. miR-149-5p was downregulated in our EC samples (2^−ΔΔCT^ = 1.64). miR-149-5p is suggested to be a valuable marker for future targeted therapy and prognosis improvement [49]. Although its role as a tumor suppressor in multiple human cancer developments has been demonstrated, including cancers of the reproductive system [49,50], fewer studies on the miR-149-5p signaling pathway in EC exist. One study showed that miR-149-5p was modulated by circular RNA (circRNA), i.e., hsa_circ_0061140, which promoted endometrial carcinoma progression by regulating the miR-149-5p/Stat3 pathway [51]. Our findings of the involvement of miR-149-5p in EC and the downregulation of miR-149-5p in EC endometrial fluid specimens are supported by the reported results [51]. A detailed summary of the references regarding the 14 miRNAs selected in the current study is presented, and their main target mRNAs and signaling pathways are listed in Appendix A [18,19,20,21,22,23,26,27,47,48,52,53,54,55,56,57,58,59,60,61].

### 3.2. Several Unique miRNAs Identified in EC Endometrial Fluid

There were several unique miRNA markers identified in EC endometrial fluids in this study. They were miR-362-3p, miR-1321, miR-4705, and miR-378c. To our knowledge, this is the first time that they have been reported in endometrial cancer. miR-362-3p was previously identified as a tumor suppressor gene, and the downregulation of miR-362-3p resulted in tumor progression, migration, and invasion by targeting SERBP1, p130Cas, and BCAP31 in ovarian, breast, and cervical cancers, respectively [62,63,64]. In contrast, we found that miR-362-3p was upregulated in EC endometrial fluid specimens (Table 1). Although the change in expression level was below the 2-fold threshold (2^−ΔΔCT^ = 1.66), its fold-change was still suggestive. The discrepancy between our findings (upregulation) and those previously reported (downregulation) could be attributed to the disparity in tissue type, i.e., EC endometrial fluid vs. ovarian, breast, and cervical specimens, and freshly collected endometrial fluid (liquid biopsy) vs. archived tissue specimens (FFPE) or cell lines. A downregulation of miRNA-1321 was identified in the EC endometrial fluid samples (2^−ΔΔCT^ = 1.96). Previously, miR-1321 was reported to be involved in ovarian cancer cell invasion and migration [65], and the downregulation of miR-1321 was reported in ovarian cancer via the upregulation of NEAT1. Herein, the downregulation of miR-1321 was identified in endometrial fluid specimens in this study. Interestingly, miR-4705 was found to be downregulated in the EC endometrial fluid samples, with a fold-change of 1.87. As a non-coding RNA, the involvement of miR-4705 in the carcinogenesis of gastric cancer was reported [66]. However, there was no previous report on miR-4705 in endometrial tissue or the involvement of miR-4705 in endometrial cancer. Another unique miRNA identified in the endometrial fluid specimens was miR-378c. It exhibited a significant downregulation (>2-fold increase in expression) in phase I of the study in 30 EC specimens. However, an opposite change in expression was detected in phase II of the study. These four microRNA biomarkers identified in the EC endometrial fluid samples have not been previously reported in the literature for endometrial cancer; thus, this study added new miRNA species to the existing panel of miRNAs [24,25,26] as potential candidate miRNA biomarkers for EC diagnosis. As a result of the criteria set initially for candidate miRNA selection from the phase I results (>2-fold change), three out of four newly identified miRNAs in the EC endometrial fluids were not selected for phase II validation. Hence, a large set of clinical samples was required to validate these miRNAs in a future study.

### 3.3. Inconsistency in miRNA Expression between Phase I and Phase II

Inconsistent results were observed in four of the fourteen miRNAs selected between phases I and phase II in the current study. Two miRNAs, i.e., miR-378c and miR-34b-3p, that were downregulated in phase I was found to be upregulated in the phase II validation. MicroRNA-378 was found to be an onco-miRNA, and its upregulation was reported in cervical cancers by targeting ST7L/Wnt/β-catenin pathways [58]. It was also reported that miR-378 inhibited proliferation and invasion in colon cancer specimens and cell lines [59], and the downregulation of miR-378 was significantly correlated with tumor growth, advanced clinical stage, metastasis, and a worse prognosis [67] in colon cancer, which helped to elucidate our finding concerning the downregulation of miR-378c (>2-fold decrease in expression), a member of the miR-378 family, which was found in the 30 EC endometrial fluid specimens in phase I. However, regarding the unique miRNA briefly mentioned in the previous paragraph, the upregulation of miR-378c was observed during phase II validation, opposite to the phase I result, with a fold-change of 1.45 averaged over 12 EC specimens. The upregulation of miR-378c obtained from the phase II results seemed to be aligned with the findings for cervical cancer [58] but differed from the findings for colon cancer [67]. Similarly, the opposite finding was also observed for miR-34b-3p in our study. The tumor suppression function of miR-34b via the mesenchymal–epithelial transition (MET) factor was demonstrated in EC tissue and a cell line by Hiroki et al. [60]. The function of miR-34b as a tumor suppressor was suggested previously. The downregulation of miR-34b in EC could be associated with cancer cell migration and invasion, which was involved in the more aggressive behavior of EC. The results from our endometrial fluid specimens from phase I are in good agreement with these findings, and the downregulation of miR-34b-5p (negative 2.25-fold) was achieved. However, this result was not validated in phase II, in which a slight upregulation (1.25-fold increase) in miR-34b-5p expression was observed. Moreover, there were two miRNAs, i.e., miR-7-5p and miR-21-5p, that were upregulated in phase I (30 EC specimens) with fold-changes of 3.3 and 2.26, respectively, but which were slightly downregulated in the phase II validation, with fold-changes of −1.14 and −1.5, respectively. In addition, the magnitude of fold-change for miR-204-5p was dramatically different in the results obtained from the two phases. In phase I, miR-204-5p showed a significant downregulation with a fold-change of 4.2 (2^−ΔΔCT^ = −4.2). However, the fold-change in the miR-204-5p downregulation was only 1.41 (2^−ΔΔCT^ = −1.41) during phase II validation. It was shown previously that miR-204-5p had a tumor suppression role in EC. It inhibited EC cell growth, migration, and invasion via the TrkB-STAT3-miR-204-5p pathway [61]. A lower expression of miR-204-5p was associated with advanced EC FIGO stages, lymph node metastasis, and a lower survival rate. This is consistent with our phase I results, in which 53% of EC patients had grade 2 cancers, and the downregulation of miR-204-5p was significant. In phase II, only 16.7% of EC patients had grade 2 histology, and a lower scale downregulation of miR-204-5p was observed.

Comparing the expression fold-changes of phase I and phase II, more than 28% of the miRNAs selected did not have a consistent correlation between the two phases (Table 2) in this study. This inconsistent direction in the expression changes was possibly related to the histopathological classification of the specimens in the two phases. In phase II, 83% of samples were grade 1, which is less aggressive than grade 2 (16.7%), while in phase I, 53% of EC samples were grade 2, and 3% were in grade 3. It appears that there was a shift in EC patients enrolled in phase II toward lower grades of endometrioid carcinoma (Table 4). Therefore, the level and the direction of miRNA expression in EC specimens may vary and be closely related to the progression of endometrial cancer. As suggested by Bao et al. [61], the regulation of some miRNAs is complex. The reduced expression of certain miRNAs could be associated with an advanced stage of EC or metastasis. An inconsistent direction of expression was documented previously for several miRNAs in endometrial cancers [25,26]. For example, miR-21 expression was reported to be upregulated in one review paper [26] but was identified to be downregulated in another [25]; both were profiled from endometrial cancer tissues. Using the same profiling methods with RT-qPCR, it was also found that both miR-204 and miR-21 exhibited different directions of expression change, as reported by different research groups [24]. Further studies with larger sample sizes are required to validate the downregulation of miR-378, miR-296-5p, miR-204-5p, and miR-34b-3p expression and upregulation of miR-7-5p and miR-21-5p expression in EC endometrial fluid specimens, to confirm the significance of their regulatory expression in EC and characterize the target signaling pathways responsibly.

### 3.4. Comparing the miRNA Expression Changes Identified in This Study with Those Reported in the Literature

Inconsistent directions of expression change for several miRNAs were observed between the findings of the current study and those reported in the literature. In addition to the two miRNAs, i.e., miR-296-5p and miR-34a-5p, as discussed in phase I results, several miRNAs also showed different directions of expression from the previously published reports. For example, miR-196a-5p was upregulated in our EC endometrial fluid specimens (fold-change = 1.81, *p* = 0.022) but downregulated in the tissue and blood samples from EC patients and patients with atypical endometrial hyperplasia (AEH) when profiled using a similar method [68]. Similarly, the expressions of miR-21-5p and miR-20a-5p were upregulated in our EC endometrial fluid samples, with fold-changes of 2.26 (*p* = 0.007) and 2.24 (*p* = 0.015), respectively, but the decreased expression of miR-21-5p and miR-20a-5p was reported by others [25]. Although a larger sample size was necessary to verify the results from the EC endometrial fluid specimens, the inconsistent direction of expression change was reviewed previously by Donkers et al. [24] for several miRNAs, including miR-21 and miR-204, which were included among our miRNA candidates. This discrepancy might be attributed to the different sample types (EC tissue (FFPE) vs. EC endometrial fluid), the tumor staging of specimens, and the methods used. Furthermore, it is commonly known that formalin fixation can cause issues with certain cellular processes and tissue autolysis. Moreover, the prolonged storage of FFPE specimens could impair RNA integrity. Additionally, during deparaffinization, the temperature and duration applied could cause DNA/RNA fragmentation and contribute to the different results achieved from archived FFPE tissue specimens versus directly collected endometrial fluid samples.

Discrepancies in the expression directions obtained in the current study and others reported in the literature could be associated with the limitations of the studies. They include the following: (1) the heterogeneity of sample types and endometrial fluid (liquid biopsy) vs. FFPE, lesion tissue, cancer cell lines, or blood samples [26]; (2) the lack of standard EC sample classification used in the reported studies—EC samples used in miRNA profiling were different in grades, types, and FIGO (International Federation of Gynecology and Obstetrics) staging [25]; (3) the sample sizes recruited in each study, which could affect the reliability of the results and the interpretation; (4) the methods used to characterize miRNAs [23] and profile their functional targets in the endometrium and EC could also contribute to a disparity in the results.

### 3.5. Clinical Implications

As previously discussed, endometrial biopsy has limits in its diagnostic capability. Transvaginal ultrasound, while useful for the risk stratification of post-menopausal patients, did not perform well for cancer detection, even in the hands of physicians experienced with ultrasound [69]. SIS is a specialized type of minimally invasive in-office ultrasound procedure during which saline is injected into the endometrial cavity to better characterize its features. The injected saline solution discarded routinely post-SIS could be collected and used as a liquid biopsy specimen for biomarker profiling. The technique evaluated in this study has the potential to offer the diagnostic capabilities of non-surgical sampling in the office setting and resource-limited settings with fewer risks.

The patient demographic features showed that age, body mass index (BMI), and contraception status were the main factors associated with endometrial carcinoma.

### 3.6. Strengths and Limitations of Our Study

This is the first study to explore miRNAs from the endometrial fluid as a potential resource for endometrial cancer detection. The study’s strengths include the following: (1) specimen collection was minimally invasive, the specimen could be collected during a routine patient in-office visit, and the technique was easy to adopt. (2) A robust exploration phase allowed for technique optimization and a reduction in chance associations. These results were confirmed using a separate, albeit limited, set of patients for validation. (3) Controls were selected from the at-risk population, which is a more suitable choice than asymptomatic controls. (4) The final group assignment during data analysis was determined by histological analysis, which is the gold standard.

We noted several limitations in the current study. First, the sample size in the validation phase was too small (22 patients). To confirm the specificity and sensitivity of EC-related miRNA identified in endometrial fluid, additional studies, ideally with a larger sample size, need to be conducted. Secondly, the histopathological classification (EC grades and type) did not match in both phases of the study. We believe this is a potential source of the low agreement in the expression levels/directions of some miRNAs obtained between phase I and phase II. For this purpose, samples could be presorted by their histology and grade prior to comparative expression runs while still maintaining lab blinding to the study group. Third, no functional targets or pathways for the unique miRNAs identified in the present study were identified and reported in this manuscript. Despite these limitations, it is promising that using the minimally invasive SIS method, we successfully collected endometrial fluid samples, isolated the RNA, and identified several miRNAs that were significantly upregulated or downregulated in EC samples.

## 4. Materials and Methods

### 4.1. Endometrial Fluid Sample Collection and Preparation

This study was conducted in the southwestern United States in a single large metropolitan community. Patient enrollment occurred at five different hospitals and clinical sites, including both public and private teaching and non-academic institutions. The study was approved by each hospital’s Institutional Review Board after review. The study population included consenting women aged 18+ who were scheduled by their gynecological provider or gynecological oncologist for hysterectomy as indicated for either endometrial cancer (including endometrial intraepithelial neoplasia) or abnormal uterine bleeding. We excluded women with a history of endometrial ablation.

Prior to the hysterectomy, endometrial fluid was collected using an intrauterine catheter to instill fluid in an identical fashion to in-office saline infusion sonohysterography (SIS). Briefly, a sterile SIS catheter (Goldstein SonoBiopsy Catheter, Cook Medical LLC, Bloomington, IN, USA) was set to a default length of 5.5 cm for acorn placement. This was passed through the external cervical os and into the uterine cavity. For endometrial fluid collection, the instilled saline was aspirated from the endometrial cavity, collected in a sterile Nunc 15 mL conical polystyrene centrifuge tube (Corning Life Science, Massachusetts, MA, USA), placed on wet ice immediately, and transported to the study laboratory within an hour of collection. For detailed information regarding endometrial fluid collection using SIS, please refer to the Appendix A for more details.

Upon arriving at the research laboratory, the sample was first centrifuged at 1900× *g* at 4 °C for 10 min using a swing bucket centrifuge (Allegra 21R centrifuge, Beckman Coulter, Indianapolis, IN, USA). The top supernatant fraction was collected and aliquoted into 2 mL sterile micro-centrifuge tubes with an attached screw cap and O-ring (Thermo Scientific, San Diego, CA, USA), with 1 mL per tube aliquot. The samples were then stored at −80 °C until they were able to be processed. After thawing on ice, second centrifugation was performed at 16,000× *g* at 4 °C for 10 min with a fixed-angle centrifuge (Eppendorf Centrifuge 5415C, Eppendorf, Enfield, CT, USA). The supernatant was then aliquoted at 200 µL per tube. One aliquot was used for miRNA isolation, and the rest of the sample aliquots were stored at −80 °C.

The volume, color, and physical properties of the endometrial fluid samples collected from the SIS procedure were carefully recorded. Bloody fluid and tissue strings were observed in some samples, which was taken into consideration during the interpretation of the results.

### 4.2. miRNA Isolation and Custom miRNA qPCR Micro-Array Plate

#### 4.2.1. Total RNA (Including miRNA) Extraction and Quantification

For RNA purification, two of the 200 µL supernatant aliquots of endometrial fluid were processed per patient sample using a miRNeasy Serum/Plasma Kit (QIAGEN, Germantown, MD, USA). The total RNA, including miRNA, was extracted according to the manufacturer’s protocol. Briefly, five volumes of QIAzol lysis reagent (1 mL) were added to a 200 µL endometrial fluid aliquot, mixed well, and incubated at room temperature for 5 min. In total, 3.5 μL of miRNeasy Serum/Plasma Spike-In Control (of 1.6 × 10^8^ copies/tube working solution) was added, followed by the addition of 200 μL of chloroform (an equal volume to the starting sample). The mixture was incubated at RT for 3 min. Centrifugation was performed at 4 °C for 15 min at 12,000× *g*. The upper aqueous phase was then transferred to a new 2 mL tube (non-screw-top tube) and the volume was measured. A 1.5× volume of 100% ethanol was added and mixed well by pipetting or inversion, 700 μL of the sample was loaded into a MinElute Spin Column, and the lid was closed. After centrifuging at room temperature for 15 s at >8000× *g*, the flow-through was discarded. This step was repeated by loading the remaining sample into the MinElute Spin Column and spinning at >8000× *g* at room temperature for 15 s. In this step, the duplicate aliquot from the same sample was loaded into the same MinElute Spin Column to increase the miRNA yield per sample for downstream analysis. The RNA was eluted in 14 µL of an elution buffer. The RNA recovery, and the cDNA synthesis efficiency were monitored via the addition of a synthetic miRNA (*C. elegans* miR-39 Spike-In Control, QIAGEN, Germantown, MD, USA) during RNA purification. The quantification of the extracted RNA samples was performed using a BioTek Epoch Microplate Spectrometer, and the RNA quality (RIN) was assessed using an Agilent 2100 Bioanalyzer and an RNA Nano Chip (Agilent Technologies, Inc., Santa Clara, CA, USA).

#### 4.2.2. First-Strand DNA Synthesis (Reverse Transcription)

Reverse transcription (RT) and cDNA synthesis were performed in a 20 µL RT reaction volume with a miScript II RT Kit (QIAGEN, Germantown, MD, USA), following the manufacturer’s protocol. The eluted RNA that was added depended on the RNA concentration, and a maximum of 12 µL could be used for one 20 µL cDNA synthesis reaction.

#### 4.2.3. Customized 96-Well miScript miRNA PCR Array Plate and Real-Time qPCR Amplification

A total of 84 candidate miRNAs were selected through a thorough search of the literature from previously published studies of miRNA species identified in EC and miRNA biomarkers reported in cancers of the female reproductive system. For custom miScript miRNA PCR Arrays (QIAGEN, Germantown, MD, USA), a 96-well micro-array plate was manufactured for a real-time qPCR array, and the wells were coated with 84 selected miRNA primers (Appendix A). Real-time qPCR was performed using a Stratagene Mx3005P instrument (Agilent, Santa Clara, CA, USA), following the manufacturer’s protocols. Taking into consideration the fact that some samples had a low concentration of extracted RNA, a pre-amplification step was applied using the QIAGEN miScript PreAmp kit (QIAGEN, Germantown, MD, USA). For proper in-plate control, each plate contained duplicates of the Spike-In cel-miR-39-3p control, housekeeping genes (SNORD61 and SNORD68), triplicates of the miRNA RT control (miRTC), and positive PCR controls (PPC). During phase I of the study, one custom miScript miRNA PCR Array plate was used for each endometrial fluid miRNA sample. A total of 60 custom miScript miRNA PCR Array plates were run. The baseline was set from cycle 2 with the “Linear View” amplification plot. The threshold was set above the background signal and within the lower half of the log-linear range of the logarithmic amplification plot.

For phase II of the study, a new custom miScript miRNA PCR Array plate was manufactured with 14 miRNAs selected from phase I results, and six sets of the samples could be run on one plate. At least one duplicate run was performed for each miRNA species.

#### 4.2.4. Data Analysis and Normalization

The resultant data from the miScript PCR Array were imported to a QIAGEN spreadsheet template and uploaded to the online analysis software (Appendix A). A global Ct mean of the miRNA targets expressed by all samples was calculated using the analysis/normalization method that was available via the manufacturer’s online service to identify the miRNA species detected and which miRNAs were up or downregulated. For the selected housekeeping genes, expression levels were not affected by disease or metabolic status. Therefore, the expression level of each of the 84 specific miRNAs (Ct values) was compared with those of the housekeeping genes on the same plate to achieve normalized Ct values, i.e., as delta Ct (ΔCt). The changes in cancer-related miRNA expression levels were calculated by comparing the average of the ΔCt values from the known cancer patients with the average of the ΔCt values obtained from the control patients for each miRNA species (ΔΔCt values) on the same plate. The ΔΔCt values obtained included the changes in expression identified in endometrial cancer patients, which could indicate positive (upregulated), negative (downregulated), or no change when compared to the controls. For each of the miRNA species identified, ΔΔΔCt values were calculated by comparing the average expression changes (ΔΔCt values) of all EC samples with the average expression levels (ΔΔCt values) of all the control samples from all plates.

A manufacturer-recommended threshold of 35 was used to determine whether any gene was expressed or not for the non-pre-amplified specimens. After normalization, fold-change (2^−ΔΔCT^) was calculated as the normalized gene expression (2^−ΔΔCT^) in the test sample divided by the normalized gene expression (2^−ΔΔCT^) in the control sample. Fold-change in miRNA biomarkers was presented for both upregulated and downregulated miRNAs relative to the controls. Both up- and downregulations were evaluated for a possible association with cancer as the final pathology-based diagnosis.

Receiver operating characteristic (ROC) curves were created with various levels of ΔΔΔCt values for miRNAs amplified from endometrial fluid specimens with the ROC analysis software, using the benign and cancer histopathology of the endometrium specimen as the gold standard. Fitted ROC curves, areas under the curve (AUCs), and standard errors were reported for prominent candidate miRNAs.

To avoid analysis bias during the validation phase, the statistician was blinded to cancer status, and an analysis was conducted for “group-1” and “group-2”, which were later recorded as the benign and malignant groups.

### 4.3. Pathological Diagnosis

Surgical specimens were processed by the hospital pathologists per the standard of care and were used as the gold standard when correlating the final pathological diagnosis. Other relevant demographic information was collected from the patient’s charts.

### 4.4. Experimental Design—Phase I (Exploration) and Phase II (Validation)

Phase I—We hypothesized that a subset of miRNAs tested in the first phase of the study (the exploration phase) could be related to a final diagnosis of endometrial cancer. This helped us to narrow down the scope of the target miRNAs to be studied in the second phase of the study (the validation phase). The proposed sample size of 60 (30 patients per group) and the analysis of 84 miRNAs allowed us to detect 2-fold differences with at least a 99% statistical power and a per-miRNA value of alpha for 0.0119 with an acceptable number of false positives of 1 and a standard deviation of 0.7 during the study’s exploration phase.

Phase II—The miRNAs with the largest absolute ΔΔCt values obtained from phase I, when compared to the controls, were considered as candidates for the validation phase of the study, with a predefined cut-off of at least a 2-fold change. When miRNA species exhibited similar fold-change values, preference was given to those that had been identified in previous studies to be associated with endometrial cancer. Up to 14 miRNA biomarkers were selected for validation based on logistical constraints, and a new custom miScript miRNA PCR Array plate was designed and manufactured for phase II of the study.

An independent set of 22 patient samples was tested on the custom miScript miRNA PCR Array plate with 14 miRNAs selected from phase I. Correlation analysis was performed on the miRNA data collected during the validation study phase to examine the association between endometrial cancer-specific miRNA expressions and clinical findings. Samples were tested in duplicate or triplicate to assess the sample and method quality.

## 5. Conclusions

Herein, we present a feasibility study that focuses on the identification of EC-specific miRNA biomarkers in an endometrial fluid using a method similar to SIS, which is a widely practiced in-office procedure. A few promising EC-related candidate miRNAs were identified in the endometrial fluid specimens, such as miR-183-5p, miR-429, and miR-146a-5p among the upregulated miRNAs and miR-296-5p and miR-204-5p among the downregulated miRNAs. Several unique miRNAs, i.e., miR-362-3p, miR-1321, miR-378c, and miR-4705, which function as tumor suppressor miRNAs in a variety of cancers, were distinguished in EC in this study.

A larger validation set of samples and the further refinement of the data analysis with standard FIGO staging will likely be required to assess each individual miRNA’s diagnostic potential as an early EC biomarker. The findings reported in this study identifying miRNAs in EC endometrial fluid samples provide a minimally invasive approach for screening patients with abnormal uterine bleeding and insights for the early diagnosis of EC.

## Figures and Tables

**Figure 1 ijms-24-08683-f001:**
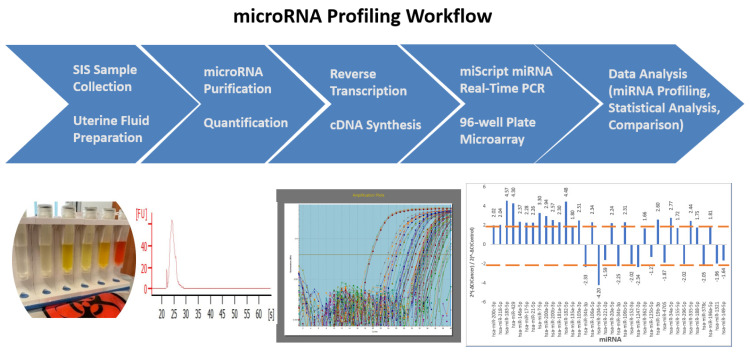
A schematic presentation of the miRNA profiling workflow applied to the endometrial fluid samples collected using sonohysterography (SIS). Five main steps are outlined, including endometrial fluid sample collection and preparation, RNA isolation and quantification, reverse transcription, and cDNA synthesis, miScript miRNA real-time PCR, and data analysis. The images shown beneath the workflow scheme are real images of the results.

**Figure 2 ijms-24-08683-f002:**
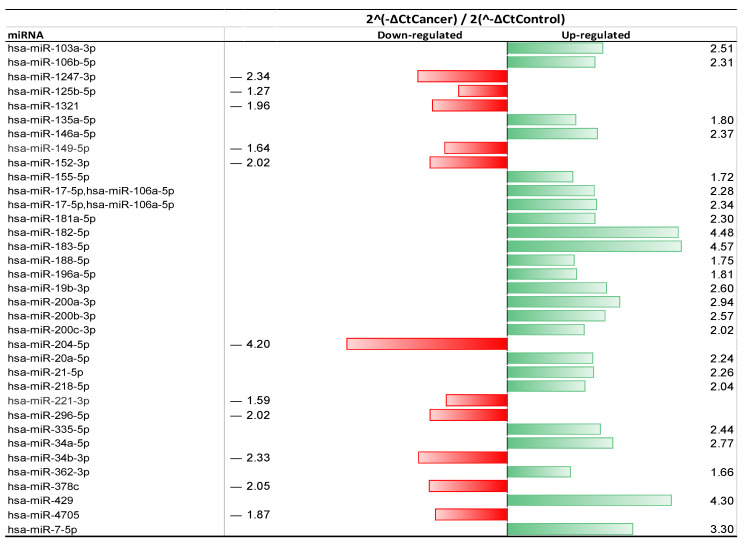
Heatmap of the fold-changes of endometrial cancer (EC)-related miRNAs. In phase I, a total of 60 patients were recruited, i.e., 30 EC patients and 30 non-cancer patients were paired. A total of 84 miRNAs were screened. In total, 25 miRNA species out of 35 identified in EC patients showed significant fold-changes, compared with non-EC patient controls. Green bars indicate fold-changes in miRNAs exhibiting upregulation. Red bars indicate fold-changes in miRNAs exhibiting downregulation.

**Figure 3 ijms-24-08683-f003:**
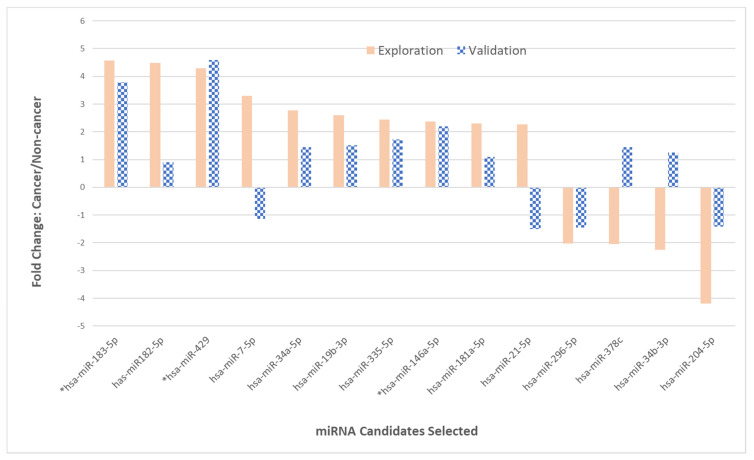
The expression levels of 14 miRNAs were compared between phase I (exploration) and phase II (validation) for EC patient endometrial fluid samples. The 14 miRNAs that showed significant changes in the expression in phase I (solid brown bars) were recruited for validation in phase II (patterned blue bars). The miRNA species with asterisks indicate the consistency of the fold-change when observed in EC samples for the two phases.

**Figure 4 ijms-24-08683-f004:**
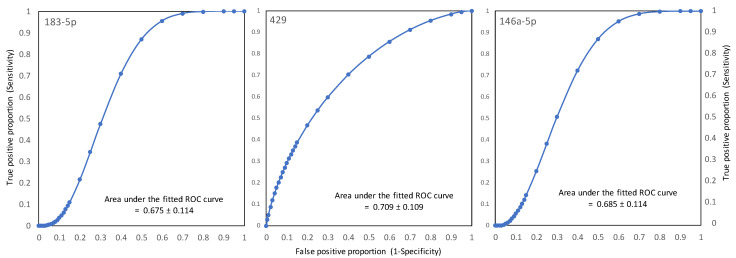
The receiver operating characteristic curves (ROC) were plotted with a fold-change in Ct values (ΔΔΔCt) at various levels of three miRNAs, i.e., miR-183-5p, miR-429, and miR-146a-5p, identified from the EC endometrial fluid specimens to differentiate between the EC and non-EC histopathology of the endometrium.

**Table 1 ijms-24-08683-t001:** Summary of miRNA expression changes identified in EC endometrial fluid specimens from phase I of the study. The fold-changes in miRNA expression identified in EC endometrial fluid specimens from phase I of the study. In the top part, all miRNAs that exhibited more than a 2-fold change in expression level (either up or downregulation) are listed. The matched miRNAs represent the miRNAs that were identified in the EC samples and previously reported in the literature. The unmatched miRNAs represent the miRNAs identified in the EC endometrial fluid samples in the current study that were not reported in EC previously based on a search of the literature. The highlighted cells are the 14 miRNAs that were selected to enter phase II of the study for validation. In the bottom part, all miRNAs identified that exhibited less than a 2-fold change are listed.

	**>2-Fold Change**	
	**Up-Regulated**	**Down-Regulated**
**Matched miRNA**	miR-182-5p	miR-204-5p
	miR-183-5p	miR-152-3p
	miR-429	miR-1247-3p
	miR-200a-3p	miR-34b-3p
	miR-200b-3p	miR-296-5p
	miR-200c-3p	
	miR-106b-5p	
	miR-34a-5p	
	miR-20a-5p	
	miR-7-5p	
	miR-218-5p	
	miR-17-5p	
	miR-106a-5p	
	miR-21-5p	
	miR-181a-5p	
	miR-103a-3p	
	miR-19b-3p	
	miR-335-5p	
	miR-146a-5p	
**Unmatched miRNA**		miR-378c
	**<2-Fold Change**	
	**Up-regulated**	**Down-regulated**
**Matched miRNA**	**miR-155-5p**	**miR-221-3p**
	miR-188-5p	miR-149-5p
	miR-196a-5p	miR-125b-5p
	miR-135a-5p	
**Unmatched miRNA**	miR-362-3p	miR-4705
		miR-1321

**Table 2 ijms-24-08683-t002:** Validation of 14 miRNAs’ expression fold-changes in endometrial fluid in phase II. The up- and/or downregulation of 14 miRNAs selected in phase I of the study was validated in phase II with another set of 22 patient samples. The asterisks indicate that the fold-change in expression levels of those miRNAs were similar in both phases.

			2^−ΔCtCancer^/2^−ΔCtControl^	
Count	Functional Change Identified	miRNA Selected	Phase I	Phase II
1	Up-regulated	* hsa-miR-183-5p	4.57	3.79
2		hsa-miR-182-5p	4.48	1.34
3		* hsa-miR-429	4.30	4.59
4		hsa-miR-7-5p	3.30	−1.14
5		hsa-miR-34a-5p	2.77	1.45
6		hsa-miR-19b-3p	2.60	1.51
7		hsa-miR-335-5p	2.44	1.73
8		* hsa-miR-146a-5p	2.37	2.19
9		has-miR-181a-5p	2.30	1.10
10		hsa-miR-21-5p	2.26	−1.50
11	Down-regulated	hsa-miR-296-5p	−2.02	−1.45
12		hsa-miR-378c	−2.05	1.45
13		hsa-miR-34b-3p	−2.25	1.25
14		hsa-miR-204-5p	−4.20	−1.41

* Indicating there was a strong correlation between Phase I and Phase II for that miRNA expressed in EC Specimens.

**Table 3 ijms-24-08683-t003:** Clinical and demographic features of patients with benign and malignant specimens. The clinical and demographic features of 82 patients enrolled in the study. Categorical variables were compared against the reference (Ref) and expressed as counts and percentages of the total numbers (n (%)). Continuous variables, such as age, BMI, gravidity, and parity, are presented as means and standard deviations. Median and data range are shown in parentheses. Ref indicates reference category used for chi-square tests.

Patient-Related Features	Benign (n = 40) *	Cancer (n = 42) *	*p* **
Categorical variables, n (%)			
Race			
White	34 (87.18)	38 (90.48)	Ref
Black	4 (10.26)	3 (7.14)	0.642
Asian	0 (0)	1 (2.38)	0.534
N. American	1 (2.56)	0 (0)	0.480
Ethnicity			
Hispanic	25 (64.1)	21 (50)	0.211
Non-Hispanic	14 (35.9)	21 (50)	Ref
Contraception			
None	11 (28.21)	5 (11.9)	Ref
Some form of contraceptives	23 (58.97)	6 (14.28)	0.456
Post-menopause	5 (12.82)	29 (69.05)	<0.001
Smoking status			
Former or current smoker	11 (28.21)	5 (11.9)	0.092
Never smoked	28(73.68)	37(88.1)	Ref
STI			
STI/history of	3 (7.69)	2 (4.76)	0.622
None	36 (92.31)	40 (95.24)	Ref
Continuous variables, mean ± sd (median, range)			
Age	45.5 ± 5.6 (45, 36–61)	58.3±12.3 (57.5, 36–88)	<0.001
BMI	32.4 ± 5.8 (31.9, 21.7–44.6)	37.8±8.9 (37.4, 20.5–56)	0.005
Gravidity	3.2 ± 1.6 (3, 0–9)	2.7±1.9 (3, 0–8)	0.209
Parity	2.6 ± 1.6 (3, 0–9)	2.2±1.9 (2, 0–9)	0.305

* Some counts may be lower due to missingness. ** Wilcoxon rank-sum test for continuous variables and mid-point exact *p*-values based on chi-square test for categorical variables.

**Table 4 ijms-24-08683-t004:** Histopathological classification of endometrial cancer specimens. Histopathological classification of the EC endometrial fluid specimens. There were 30 EC case samples and 12 EC case samples in phase I and phase II of this study, respectively. The cancer grade classification was based on pathology reports. EIN—endometrial intraepithelial neoplasia.

		Phase I		Phase II	
Histologic Type	Grade	# of Cases	% of Total	# of Cases	% of Total
EIN *****	-	4	13.33	0	0
Endometrioid		25	83.33	12	100
	1	8	27	10	83.33
	2	16	53	2	16.66
	3	1	3.33	0	0
Rhabdomyosarcoma		1		0	
	3	1	3.33	0	0
**Total**		30	100	12	100

***** Endometrial Intraepithelial Neoplasia.

## Data Availability

Data can be available upon request, due to the HIPA security information.

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
