# Peer review of "Identification of Endometrial Cancer-Specific microRNA Biomarkers in Endometrial Fluid"

_ijms, 2023, doi:10.3390/ijms24108683_

Round 1
Reviewer 1 Report
The authors developed a method for the evaluation of miRNA levels from endometrial fluids for the early detection of endometrical cancer. Specimen collection is less invasive compared to the traditional tissue biopsy approach. There is novelty in the authors’ approach. However, their correlation between miRNA levels and endometrical cancer was not conclusive due to the inherent specimen variation and the limited number of specimens used in the validation studies.
The manuscript will benefit from language re-editing. Furthermore, the manuscript could be more concise. The discussion section has a lot of repetition from the introduction section. The discussion section should focus more on highlighting the strengths and limitations of the study. Below are additional comments that the authors should address.
1. If this method is adopted in clinical setting, what will be the total turnaround time to get the patient results?
2. Are there any data from corresponding biopsy tumor tissues from endometrial cancer patients? miRNA profile from tumour biopsy vs endometrical fluid samples would provide further insight about the utility of the developed method
3. Given the fact the the developed method is prone to specimen derived inhibition, are there other methods for miRNA detection that the authors could have used instead? If so, this should be discussed in the discussion section.
The manuscript will benefit from language re-editing. Furthermore, the manuscript could be more concise. The discussion section has a lot of repetition from the introduction section
Author Response
- If this method is adopted in clinical setting, what will be the total turnaround time to get the patient results?
Response 1: The collection of endometrial fluid with SIS method takes 5 min at the bedside. The specimen collected is placed on ice immediately and transferred to the lab where the sample is processed. It takes 1-2 days from specimen collection to miRNA profiling.
In comparison, office EMB is collected at the bedside and immediately placed in formalin. The sample is then transferred to a lab where the sample is processed, and slides are prepared for a pathologist’s review. In some cases, special tissue stains are warranted which take additional time to prepare. Under typical circumstances, results return within about a week.
- Are there any data from corresponding biopsy tumor tissues from endometrial cancer patients? miRNA profile from tumour biopsy vs endometrical fluid samples would provide further insight about the utility of the developed method
Response 2: Great point.
We did not have data from corresponding biopsy tumor tissues of EC patients in this study. Considering some differences found in the results obtained from this study and other published from FFPE or biopsies, it is important and necessary to design and conduct a study, comparing the results of EC endometrial fluid with corresponding biopsied tumor tissue from the same patient, to provide further insight about the utility of the developed method.
However, as we suggested in discussion, the inconsistent results in miRNA expression between the published paper and our findings OR between the two phases of our studies were clearly attributed to not only the sample types (liquid biopsy vs tumor tissue), sample collection location (biopsied site), but also the FIGO classification of the EC specimen. The endometrial fluid collected using the current method (SIS technique) was minimally invasive, can be performed during in-office visit, and the biomarkers discovered might have the benefit of early diagnosis. The biomarkers identified from tumor tissue or FFPE were mostly the miRNA biomarkers for late-stage EC or mixed stages depending on the biopsied site and was post-operative in nature. Therefore, the expression in miRNA biomarkers might not correlate well between the endometrial fluid and tumor tissue biopsied, even for the same patient.
- Given the fact the the developed method is prone to specimen derived inhibition, are there other methods for miRNA detection that the authors could have used instead? If so, this should be discussed in the discussion section.
Response 3: Thank you for the comments. There are many different approaches for miRNA detection, including microarrays, NGS, and qRT-PCR etc. The most sensitive, accurate and popular one is the qRT-PCR in a microarray format. Regardless of what method is used for detection, PCR is the step prone to inhibition. Therefore, the SIS technique developed for endometrial fluid collection itself is not prone to specimen derived inhibition.
The quality of RNA isolated from the clinical specimen (body fluid or tumor tissue) is critical for any downstream applications in miRNA research. There are many approaches people used for miRNA isolation, for example, ultracentrifugation (UC), size exclusion chromatography (SEC), bead-based isolation, and many other commercial kits. The UC gives high purity of miRNA, but is very labor-intensive, time consuming, requires a large sample volume, and generally has a lower yield. The SEC is often used for exosome isolation. We selected QIAGEN miRNeasy kit which appeared most often in the literature at that time, and they provided the customized miRNA RT-qPCR microarray plates we used to screen our specimens. Since optimization of sample prep for the endometrial fluid specimen was not our goal in this study, we didn’t try other methods for miRNA isolation and detection.
As we mentioned in the discussion, the major contributors to the variation in miRNA quantification and PCR inhibition came from the specimen due to variations in the pathophysiological conditions of the endometrial fluid (such as contaminants from non-endometrial epithelium or other non-tumor sources). The encountered PCR inhibition occurred about 6% of the time in our 82 patient specimens, and usually repeating the miRNA isolation with another aliquot of the same patient specimen would resolve the issue. We can elaborate more in the discussion section.
Comments on the Quality of English Language The manuscript will benefit from language re-editing. Furthermore, the manuscript could be more concise. The discussion section has a lot of repetition from the introduction section
Response: Thank you, the reviewer for taking time and patience reviewing our manuscript. Your valuable comments are well-taken. We took your advice and used English editing service to improve the fluency and quality of the manuscript language. We tried to be more concise in the discussion section and reduced the repetition.
Reviewer 2 Report
The manuscript reports on the development of a method to detect microRNA biomarkers specific to endometrial cancer (EC) from liquid biopsy samples of endometrial fluid, collected during patient-scheduled in-office visits using the same technique performed for saline infusion sonohysterography (SIS). The study was conducted in two phases, an exploratory phase I and a validation phase II. A total of 82 endometrial fluid samples were collected and processed, and 14 microRNA biomarkers with the greatest expression variation from phase I were selected for phase II validation and statistical analysis. The study identified several promising candidate microRNAs in the endometrial fluid samples, such as miR-183-5p, miR-429, miR-146a-5p of upregulated microRNAs and miR-296-5p and miR-204-5p of downregulated microRNAs. In addition, several unique microRNAs that function as tumor suppressor microRNAs in a variety of cancers were distinguished in EC.
Comments for authors:
1. Only 82 patients' endometrial fluid samples were obtained and analyzed, which is a tiny sample size I am fully aware of the difficulties and the significance of clinical samples. The sample size of 60 healthy controls and 22 patients with endometrial carcinoma may not be large enough to draw firm conclusions about the value of microRNA biomarkers for EC diagnosis
2. Inadequate statistical analysis was included in the submitted publication. Additional statistical analysis is needed to support the claims that the identified microRNAs had a consistent and large fold change in upregulation and that four miRNAs were uniquely found. To better illustrate the importance and extent of the differences found, the authors should offer p-values, confidence ranges, and effect sizes.
3. Introduction This article gives a helpful overview of the difficulties in diagnosing endometrial cancer now, as well as the promise of liquid biopsy as a diagnostic tool. A more thorough review of the literature on liquid biopsy for endometrial cancer detection, including the benefits and drawbacks of different types of liquid biopsies and the numerous biomarkers that have been explored, would be helpful.
4. A more in-depth explanation of microRNAs, particularly their function in carcinogenesis and their potential as biomarkers, would enrich this section. Additional validation studies may be required, and there is always the chance of false positives and negatives, all of which should be considered in this section on the drawbacks of employing microRNAs as biomarkers.
5. The objective of the study. It would be helpful to provide a clear research question or hypothesis to guide the reader through the study.
6. The authors have not provided enough details on the research process. The authors should elaborate on the SIS technique and its implementation in order to collect endometrial fluid samples. There should also be additional detail about the methods used to extract RNA and profile microRNAs.
7. Authors should conduct a larger validation study using a larger sample size to confirm the potential of the identified biomarkers for early diagnosis of EC.
8. The clinical utility of individual microRNAs can be determined by comparing their diagnostic potential with the usual FIGO staging, which the authors should utilize.
9. Authors should provide a clear discussion of the clinical implications of their findings, including how the proposed method might improve the diagnosis and treatment of EC in clinical practice.
10. Authors should provide a clear description of the limitations of their study and provide suggestions for future research in the field. although the level of caution presented in the manuscript is appropriate given the complexity of developing biomarkers for cancer diagnosis. however a bit of more explanation i feel is needed
Author Response
- Only 82 patients' endometrial fluid samples were obtained and analyzed, which is a tiny sample size I am fully aware of the difficulties and the significance of clinical samples. The sample size of 60 healthy controls and 22 patients with endometrial carcinoma may not be large enough to draw firm conclusions about the value of microRNA biomarkers for EC diagnosis
Response 1: Thank you for pointing out the difficulties and significance of obtaining clinical samples. A total of 82 patients were recruited in this feasibility study, with 60 patients in phase I and 22 patients in phase II. In phase I, there were 30 controls and 30 EC patients. In phase II, there were 10 controls and 12 EC patients. The sample size of 40 controls and 42 EC patient specimens was reported in this study. We agree with the reviewer’s comment that the sample size may not be large enough to draw firm conclusion, which we emphasized in the manuscript. In the manuscript, we also highlight the value of endometrial fluid as the proper clinical specimen in discovery of molecular biomarkers for EC early diagnosis, versus FFPE, tumor-derived cell lines or patient blood samples.
- Inadequate statistical analysis was included in the submitted publication. Additional statistical analysis is needed to support the claims that the identified microRNAs had a consistent and large fold change in upregulation and that four miRNAs were uniquely found. To better illustrate the importance and extent of the differences found, the authors should offer p-values, confidence ranges, and effect sizes.
Response 2: Supplemental table of fold-change analysis (Appendix S3_Phase I miRNA Expression Fold-change Analysis) has been submitted with the revised manuscript, to further support the Table 1 results. P-values and effect size (magnitude of fold change) have been included in the table. Footnotes have been added to explain the results to improve the clarity of the presentation.
- Introduction This article gives a helpful overview of the difficulties in diagnosing endometrial cancer now, as well as the promise of liquid biopsy as a diagnostic tool. A more thorough review of the literature on liquid biopsy for endometrial cancer detection, including the benefits and drawbacks of different types of liquid biopsies and the numerous biomarkers that have been explored, would be helpful.
Response 3: Thank you for the comment. There are a limited number of reports in the literature on using liquid biopsy samples for EC detection. As we mentioned in the introduction and discussion, the only published studies available in the literature focused on either ovarian cancer biomarkers or EC protein markers using cervicovaginal lavage or uterine lavage specimens, while we reported here to identify miRNA biomarkers in EC endometrial fluid specimens. A review article published recently by Shen et al. (Ref. [8] in the manuscript) supports our study on minimally invasive approaches for early EC detection. In the two studies mentioned in this review, the uterine lavage specimens from 5 and 7 EC cases, respectively, were reported. Currently, there is no standardization in biomarker identification from liquid biopsy samples, and the drawback is the lack of consistency in the applied methods used for miRNA biomarker characterization.
As for the pros and cons of different types of liquid biopsies and numerous biomarkers explored in the literature, that would be a huge topic to discuss and hard to have a complete review in the current scope of work.
- A more in-depth explanation of microRNAs, particularly their function in carcinogenesis and their potential as biomarkers, would enrich this section. Additional validation studies may be required, and there is always the chance of false positives and negatives, all of which should be considered in this section on the drawbacks of employing microRNAs as biomarkers.
Response 4: Thank you for the comment. Most of the miRNA markers discussed in the current study were selected from the published studies in literature and we, herein, were trying an alternate, and arguably more appropriate, method to identify them in the liquid biopsy samples. A couple of review papers on miRNA biomarkers in EC and their functions in carcinogenesis were cited in the manuscript.
We totally agree that additional validation studies are required to confirm the findings. As we mentioned in the discussion section, the inconsistent results (miRNA expression) were not only reported in the literature from EC FFPE or biopsy specimens due to the sampling biases, but also obtained in the current study from the same sample type, likely due to different EC tumor staging in each group. This was why we suggested 1) for the early detection, the endometrial fluid collected with this minimally invasive method is preferrable to FFPE or biopsy specimens for EC miRNA biomarker identification, and 2) The EC tumor FIGO staging is important in identifying EC miRNA biomarkers for diagnosis.
- The objective of the study. It would be helpful to provide a clear research question or hypothesis to guide the reader through the study.
Response 5: Yes, we can revise that as the following:
The objective of this study was to perform a proof-of-concept, develop a minimally invasive way to collect endometrial fluid during patient in-office visit, and to identify and validate miRNA biomarkers which could differentiate benign from cancerous endometrial conditions.
In future studies to validate the miRNA biomarkers identified in EC endometrial fluid with a larger set of samples, we will have more specific questions and hypotheses in their expression regulations and signaling pathways.
- The authors have not provided enough details on the research process. The authors should elaborate on the SIS technique and its implementation in order to collect endometrial fluid samples. There should also be additional detail about the methods used to extract RNA and profile microRNAs.
Response 6: We considered that the details of collection technique to be more of a clinical procedure, and therefore, we left the details in the supplementary materials – please see Appendix S1 – Method.
The details of method on RNA isolation were outlined step-by-step in the methods, section of miRNA isolation and custom miRNA qPCR microarray plate for miRNA profiling, using QIAGEN’s products.
- Authors should conduct a larger validation study using a larger sample size to confirm the potential of the identified biomarkers for early diagnosis of EC.
Response 7: Absolutely agreed. We are going to do that in a future study.
- The clinical utility of individual microRNAs can be determined by comparing their diagnostic potential with the usual FIGO staging, which the authors should utilize.
Response 8: Great question. At this stage, both ROC and FIGO should be considered. As we discussed in the manuscript, an inconsistent result in terms of miRNA expression was obtained in our phase I and phase II studies. We found that it was mainly attributed by histopathological classification of EC patients, early stages vs. late stages, in each cohort. … That was why we suggested that the community should use the same FIGO to standardize the classification of EC specimen, during EC biomarker discovery, followed by validating the diagnostic potential using ROC analysis with FIGO classified specimens before its clinical implementation as a biomarker assay for cancer diagnosis.
- Authors should provide a clear discussion of the clinical implications of their findings, including how the proposed method might improve the diagnosis and treatment of EC in clinical practice.
Response 9: Excellent suggestion. While the proposed technique still has room for more vetting, an improvement in EC diagnostics is needed to improve time to diagnosis and appropriate cancer staging, particularly in resource limited locations. Research estimates on the accuracy and tissue sampling of endometrial biopsy indicate that it samples around 4% of the endometrial surface (1), has a sensitivity of around 83% (2), and its agreement with final pathology is around 73% (3). The technique, herein studied, when optimized has the potential to sample a greater percentage of the endometrial surface and should have an associated increase in these accuracies. Endometrial biopsy also relies on the availability of a pathologist to review the tissue specimen. Our technique can be performed by a laboratory technician. If proven to be reliable to tumor grading and histology, the results could also be used to triage lower risk patients to the surgical care of a general gynecologist without compromising treatment success.
- Rodriguez GC, Yaqub N, King ME. A comparison of the Pipelle device and the Vabra aspirator as measured by endometrial denudation in hysterectomy specimens: the Pipelle device samples significantly less of the endometrial surface than the Vabra aspirator. Am J Obstet Gynecol. 1993;168(1 Pt 1):55-59. doi:10.1016/s0002-9378(12)90884-4
- Guido RS, Kanbour-Shakir A, Rulin MC, Christopherson WA. Pipelle endometrial sampling. Sensitivity in the detection of endometrial cancer. J Reprod Med. 1995;40(8):553-555.
- Visser NCM, Reijnen C, Massuger LFAG, Nagtegaal ID, Bulten J, Pijnenborg JMA. Accuracy of Endometrial Sampling in Endometrial Carcinoma: A Systematic Review and Meta-analysis. Obstet Gynecol. 2017;130(4):803-813. doi:10.1097/AOG.0000000000002261
10. Authors should provide a clear description of the limitations of their study and provide suggestions for future research in the field. although the level of caution presented in the manuscript is appropriate given the complexity of developing biomarkers for cancer diagnosis. however a bit of more explanation i feel is needed
Response 10: Thank you and we will elaborate more in the “Strength and limitation of our study” in the discussion section.
Round 2
Reviewer 1 Report
No further comments